# Transient Receptor Potential Ankyrin 1 (TRPA1) Channel Mediates Acrolein Cytotoxicity in Human Lung Cancer Cells

**DOI:** 10.3390/ijms241411847

**Published:** 2023-07-24

**Authors:** Akihiko Sakamoto, Yusuke Terui, Kazuei Igarashi, Keiko Kashiwagi

**Affiliations:** 1Faculty of Pharmacy, Chiba Institute of Science, Choshi 288-0025, Japan; asakamoto@cis.ac.jp; 2Amine Pharma Research Institute, Innovation Plaza at Chiba University, Chiba 260-0856, Japan; iga16077@gmail.com

**Keywords:** acrolein, transient receptor potential ankyrin 1 (TRPA1), cell damage, cytotoxicity

## Abstract

Transient receptor potential ankyrin 1 (TRPA1) is a nonselective ion channel implicated in thermosensation and inflammatory pain. It has been reported that expression of the TRPA1 channel is induced by cigarette smoke extract. Acrolein found in cigarette smoke is highly toxic and known as an agonist of the TRPA1 channel. However, the role of TRPA1 in the cytotoxicity of acrolein remains unclear. Here, we investigated whether the TRPA1 channel is involved in the cytotoxicity of acrolein in human lung cancer A549 cells. The IC_50_ of acrolein in A549 cells was 25 μM, and acrolein toxicity increased in a concentration- and time-dependent manner. When the effect of acrolein on TRPA1 expression was examined, the expression of TRPA1 in A549 cells was increased by treatment with 50 μM acrolein for 24 h or 500 μM acrolein for 30 min. AP-1, a transcription factor, was activated in the cells treated with 50 μM acrolein for 24 h, while induction of NF-κB and HIF-1α was observed in the cells treated with 500 μM acrolein for 30 min. These results suggest that acrolein induces TRPA1 expression by activating these transcription factors. Overexpression of TRPA1 in A549 cells increased acrolein sensitivity and the level of protein-conjugated acrolein (PC-Acro), while knockdown of TRPA1 in A549 cells or treatment with a TRPA1 antagonist caused tolerance to acrolein. These findings suggest that acrolein induces the TRPA1 channel and that an increase in TRPA1 expression promotes the cytotoxicity of acrolein.

## 1. Introduction

Transient receptor potential ankyrin 1 (TRPA1), a member of the transient receptor potential (TRP) channel superfamily, is a non-selective transmembrane cation channel that is mainly involved in Ca^2+^ permeability [1]. In humans, the *TRPA1* gene is located in chromosome 8, band q21.11, comprises 73,635 bases and 29 exons, and encodes a large protein, consisting of 1119 amino acids. TRPA1 has a transmembrane core conserved among the members of the TRP channel family, consisting of six transmembrane α-helices (TM1–6) with a reentrant pore loop between TM5 and TM6. These two TM domains converge and form the central cavity of the channel, with two gates or restriction points. It is characterized by the presence of around 17 ankyrin repeats in its N-terminus [2]. The TRPA1 channel was thought to be predominately expressed in neuronal cells, but it has been reported that TRPA1 is also present in non-neuronal cells such as melanocytes, lung fibroblasts, and epithelial cells [3,4]. TRPA1 is known to be activated as a sensor by exogenous irritants but also by endogenous mediators, such as reactive oxygen (ROS) and nitrogen (RNS) species. Channel gating of TRPA1 by electrophiles occurs by covalent modification of cysteine and lysine residues within the N-terminus [5,6,7]. TRPA1 activated by these mediators causes an influx of Ca^2+^ into the cells, which has been shown to be involved in biological processes as diverse as gene expression, secretion of bioactive substances, and cell death [8,9]. TRPA1 has been reported to be associated with a number of inflammatory diseases, such as asthma, chronic obstructive pulmonary disease (COPD), and neurodegenerative disorders (multiple sclerosis, Alzheimer’s and Parkinson’s diseases) [10]. Since Ca^2+^-dependent signaling molecules play an important role in the regulation of proliferation, apoptosis, and cellular differentiation, the involvement of TRP channels has been reported in cancer [11]. Interestingly, some members of the TRP family have been found to be involved in tumor-promoting processes, whereas other TRP channels have been linked with the suppression of tumor growth. With respect to the TRPA1 channel, its expression is downregulated in tumor cells, suggesting that this protein is associated with a tumor suppressor function [12]. However, since function of the TRPA1 channel was found to be an important chemosensor for pain-eliciting substances or potentially harmful irritants, it was also found that TRPA1 may exert tumor-promoting effects. Accordingly, regulation of TRPA1 might be a target of chemotherapeutic intervention, so more basic research on TRPA1 is required to understand this regulation.

Acrolein (2-propenal, CH_2_=CH-CHO), an α,β-unsaturated aldehyde associated with oxidative stress, is formed when organic matter (including cigarette smoke, fossil fuels, and cooking oils) is combusted. In cigarette smoke, the acrolein concentration is, of course, determined by the smoking conditions, puff volume, puff rate, the type and brand of cigarettes, the method of their manufacture, and composition, but it was reported that the concentrations of acrolein in the cigarette smoke of various cigarettes ranged from 24.9 to 223 μg/cigarette. Acrolein is considered one of cigarette smoke’s most toxic and harmful components [13]. Acrolein can also be formed endogenously during cellular metabolism by (1) degradation of threonine by neutrophil-derived myeloperoxidase; and (2) amine oxidase-mediated catabolism of polyamines such as spermine and spermidine. Acrolein is known to rapidly form conjugates with cellular thiol molecules such as glutathione, cysteine, and/or thioredoxin. Among these, acrolein is metabolized into 3-hydroxypropyl mercapturic acid (3-HPMA) after conjugation with glutathione, and 3-HPMA is excreted into urine [14]. If acrolein cannot be detoxified in cells, it causes cellular damage. In contrast to ROS, such as superoxide anion (O_2_^−^), hydrogen peroxide (H_2_O_2_), and hydroxyl radical (•OH), which mainly cause DNA damage, acrolein causes cell damage by interaction with proteins involved in diverse cellular processes [15,16,17,18,19]. We previously identified acrolein-conjugated amino acids in several proteins and demonstrated the mechanism of acrolein toxicity. We first found that acrolein-conjugated glyceraldehyde-3-phosphate dehydrogenase (GAPDH) translocated to the nucleus and caused apoptosis in mouse mammary carcinoma FM3A cells [20]. It was also found that a dysfunction of the cytoskeleton induced by acrolein was strongly involved in tissue damage and dendritic spine extension [21,22]. However, acrolein-conjugated proteins are not always inactivated. For example, the activities of matrix metalloproteinase-9 (MMP-9) and proheparanase (proHPSE) are enhanced, while the antibody-recognizing abilities of immunoglobulins are modified by acrolein conjugation [19]. Acrolein has been reported to be pathologically associated with several oxidative stress-related diseases, including stroke, cancer, and Alzheimer’s disease [15,18,19]. Especially regarding stroke, the level of protein-conjugated acrolein (PC-Acro) in plasma was increased, and the multiplied value of spermine oxidase (SMOX) and acetylpolyamine oxidase (PAOX) was nearly parallel with the size of brain infarction [23]. Thus, it has been suggested that acrolein in damaged tissue is produced mainly from polyamines, especially from spermine.

TRPA1 is known to be induced and activated by cigarette smoke extract [24,25]. Acrolein is also known as a compound that activates TRPA1; however, the involvement of TRPA1 in acrolein toxicity has not been clarified in detail. In this study, we investigated whether TRPA1 expression was increased in A549 human lung cells by acrolein exposure and the mechanism responsible was explored. In addition, we also investigated whether TRPA1 mediates acrolein toxicity.

## 2. Results

### 2.1. Effect of Acrolein on Viability of A549 Cells

There are few reports evaluating the toxicity of acrolein by direct addition to cells. To identify an appropriate concentration of acrolein, the IC_50_ value for acrolein was determined by incubating A549 cells for 72 h. The viability of A549 cells against acrolein was analyzed using a CCK-8 kit. The IC_50_ value for acrolein was 25 μM in A549 cells (Figure 1A). As shown in Figure 1B, cell viability was examined by exposure to high concentrations of acrolein. After seeding A549 cells into the wells of a 96-well plate, different concentrations of acrolein were added and incubated for different times. Acrolein reduced the viability of A549 cells in a time- and concentration-dependent manner. These results suggested that acrolein is strongly cytotoxic to A549 cells.

### 2.2. Induction of TRPA1 Expression by Acrolein in A549 Cells

It has been reported that cigarette smoke extract containing acrolein induces TRPA1 expression [24,26]. Therefore, we analyzed whether acrolein induces TRPA1 expression in A549 cells using western blot analysis and qRT-PCR. Membrane fractions were extracted from A549 cells under different treatment conditions with acrolein and used for protein detection. As shown in Figure 2A, the protein level of TRPA1 was clearly increased by treatment with 50 μM acrolein for 24 h or 500 μM acrolein for 30 min. The level of TRPA1 was also increased by treatment with 100 μM acrolein for 30 min. The protein level of a membrane protein Na^+^/K^+^-ATPase, used as a control, did not change with or without acrolein. Under these conditions, the expression of *TRPA1* mRNA was measured using qRT-PCR. The level of TRPA1 mRNA was also increased to a similar extent as that of TRPA1 protein (Figure 2B). Thus, the results suggested that acrolein induces TRPA1 expression at the transcriptional level.

To investigate the mechanism of TRPA1 induction by acrolein, we searched for transcription factors in the *TRPA1* promoter region. When the “Eukaryotic Promoter Database” (EPD; https://epd.epfl.ch//index.php accessed on 31 January 2020) was used to search for transcription factors, AP-1 (activator protein 1), NF-κB (nuclear factor-kappa B), and HIF-1α (hypoxia-inducible factor-1 alpha) were found at the *TRPA1* promoter (Figure 3A). It is well known that these transcription factors are activated by cell damage, inflammation, and hypoxic stress [27]. Nuclear fractions were extracted from acrolein-treated A549 cells, and changes in the levels of these transcription factors due to acrolein exposure were assessed by western blot analysis (Figure 3B). Although the level of c-Jun, a component of AP-1, did not change by treatment with 50 μM acrolein for 24 h or 500 μM acrolein for 30 min, the level of phosphorylated c-Jun (phospho-c-Jun) was increased by treatment with 50 μM acrolein for 24 h, indicating that AP-1 was activated under these conditions. On the other hand, increases in NF-κB p65 and HIF-1α levels in nuclear fractions were observed after treatment with 500 μM acrolein for 30 min. Under these conditions, increased phosphorylation of NF-κB (phospho-NF-κB) was also observed. The protein level of nuclear protein histone H3, used as a control, did not change with or without acrolein. These results indicated that acrolein activates various transcription factors and induces TRPA1 expression.

### 2.3. Contribution of TRPA1 to Acrolein Toxicity

We next examined the contribution of TRPA1 to acrolein toxicity using a TRPA1 antagonist, HC-030031. HC-030031 is a substituted theophylline derivative which acts as a potent and selective TRPA1 inhibitor. As shown in Figure 4A, the viability of A549 cells was approximately 40% after treatment with 500 μM acrolein for 30 min, but pretreatment with HC-030031 (100 μM) significantly reduced the decrease in cell viability induced by acrolein. The result suggested that TRPA1 is associated with acrolein toxicity. To confirm the contribution of TRPA1 to acrolein toxicity, cell viability was assessed by knocking down *TRPA1* mRNA by shRNA or overexpressing TRPA1 protein by pCMV3-TRPA1 plasmids in A549 cells. The protein levels of TRPA1 in the membrane fraction became 30% of those in the control after knockdown by shRNA and were 5-fold higher than those in the control after overexpression by plasmids (Figure 4B). The protein levels of Na^+^/K^+^-ATPase as a control did not change after transfection. When these cells were treated with 500 μM acrolein for 30 min, knocking down of TRPA1 in A549 cells decreased the sensitivity to acrolein, which was similar to treatment with HC-030031. In contrast, overexpression of TRPA1 in A549 cells increased the sensitivity to acrolein (Figure 4C). These results suggested that TRPA1 promotes acrolein toxicity.

It has been reported that acrolein is mainly produced from polyamines and causes cell damage by interaction with proteins [18,19]. Acrolein rapidly reacts with lysine residues in proteins to form protein-conjugated acrolein (PC-Acro). To investigate the effect of TRPA1 on acrolein toxicity, the levels of PC-Acro and polyamines were measured. When the cells were treated with 500 μM acrolein for 30 min, the level of PC-Acro, especially the protein polymerized through crosslinking by acrolein, was increased (shown in square brackets in Figure 5A) and the polyamine content was decreased (Figure 5B). In TRPA1 overexpression cells, the level of PC-Acro was increased further, although the polyamine content was almost the same as the vector. It is suggested that acrolein can be released into cells through the TRPA1 channel [28] and conjugates with various proteins, resulting in cell damage. These results indicated that TRPA1 mediates acrolein toxicity.

## 3. Discussion

Previous studies have reported that TRPA1 is induced by cigarette smoke and contributes to the inflammatory response [24]. Cigarette smoking is a risk factor for lung cancer, contributing to lung cancer progression and metastasis. Moreover, cigarette smoking correlates with increased metastasis frequency of breast, pancreatic, and bladder cancers [29]. Involvement of the TRPA1 channel has been reported in cancer. In the present study, we focused on acrolein, the major electrophile in cigarette smoke, and we found that acrolein increased TRPA1 expression in A549 cells. Then, we verified the mechanism of TRPA1 induction by acrolein and demonstrated the involvement of TRPA1 in acrolein-mediated cell damage.

TRPA1 is expressed and functions in neuronal cells, but a few reports have shown that its expression is also induced in non-neuronal cells, including lung cancer cell lines and bronchial epithelial cells, by cigarette smoke extract (CSE) [24,26]. We found that acrolein induced the expression of TRPA1 in A549 cells (Figure 2). The results suggest that acrolein is one of the factors responsible for the induction of TRPA1 expression by CSE. We further analyzed the mechanism by which acrolein induces TRPA1 expression and found that several transcription factors were activated by acrolein. Interestingly, the transcription factors activated by acrolein differed depending on the exposure conditions (Figure 3B). The binding sites for these transcription factors were found in the promoter region of the *TRPA1* gene. In particular, it has been reported that both NF-κB and HIF-1α can bind to specific sites on the *TRPA1* gene [26,30]. We previously reported that acrolein activated AP-1 and NF-κB in mouse neuroblastoma Neuro-2a cells [31], and HIF-1α has also been reported to be induced by acrolein [32]. Our findings suggest that acrolein induces TRPA1 expression by activating several transcription factors. Recruitment and activation of immune cells is modulated by cytokines and chemokines, which are regulated by transcription factors such as AP-1 and NF-κB [33]. Phosphorylation of these transcription factors was increased by treatment with acrolein, suggesting that the signaling transduction pathway involved in inflammation was activated. HIF-1α is a protein ubiquitously expressed and notably produced by tumor cells under hypoxic conditions. HIF-1 is involved in the activation of genes involved in cancer proliferation, bone metastasis, and other associated pathways [34]. Thus, it is well known that these transcription factors play a pivotal role in inflammatory diseases and cancer. Furthermore, it has been reported that the expression and membrane translocation of TRPA1 was induced by inflammatory cytokines in A549 cells [35,36]. Acrolein naturally induces an inflammatory response [37]. These reports suggest that TRPA1 expression may be induced by the inflammatory response caused by acrolein.

It has been reported that genetic deletion of the *TRPA1* gene in mice eliminated acrolein-induced Ca^2+^ influx in trigeminal cultures [28]. We previously reported that acrolein is produced from polyamines by Ca^2+^ influx via the NMDA receptor. The influxed Ca^2+^ releases spermine from ribosomes, spermine is oxidized by spermine oxidase, and acrolein is produced [38]. The results indicate that one mode of toxicity of Ca^2+^ is correlated with increased acrolein exposure. Indeed, we showed that knockdown of TRPA1 in A549 cells or treatment with a TRPA1 antagonist attenuated acrolein toxicity, whereas overexpression of TRPA1 increased acrolein sensitivity (Figure 4). Activation of TRPA1 by acrolein increased the level of PC-Acro (Figure 5). Thus, the results indicate that TRPA1 contributes to the cytotoxicity of acrolein. However, the effect of TRPA1 on acrolein toxicity in A549 cells was not drastic. Since the sensitivity to acrolein and expression of TRPA1 may vary by cell type, the contribution of TRPA1 to acrolein toxicity may be clarified if other cell lines are used, such as neuronal cells and bronchial epithelial cells. On the other hand, it has been suggested that cellular and inflammatory responses differ depending on the conditions of acrolein exposure [37]. In fact, the transcription factors that were activated differed depending on the acrolein exposure conditions (Figure 3B). It has also been reported that TRPA1 promotes resistance to oxidative stress [39]. Although this study demonstrated the function of TRPA1 in acute high-dose acrolein exposure, TRPA1 may also function to attenuate acrolein toxicity under some conditions.

As we have discussed, TRPA1 and acrolein have been reported to be involved in smoking, inflammation, and cancer. Since this study revealed that TRPA1 mediates acrolein toxicity, TRPA1 may be a target in therapeutic strategies for several diseases in which acrolein plays a role.

## 4. Materials and Methods

### 4.1. Cell Culture

Human A549 lung cancer cells (European Collection of Authenticated Cell Cultures, Salisbury, UK) were cultured in Dulbecco’s Modified Eagle’s Medium (D-MEM) (FUJIFILM Wako Chemicals, Osaka, Japan) supplemented with 10% heat-inactivated fetal bovine serum (FBS) (Gibco/Life Technologies, Carlsbad, CA, USA) at 37 °C in an atmosphere of 5% CO_2_ in air. During the experiments, the cells were cultured with acrolein (Tokyo Chemical Industry Co., Ltd., Tokyo, Japan) and TRPA1 antagonist HC-030031 (FUJIFILM Wako Chemicals, Osaka, Japan).

### 4.2. Cell Viability Assay

About 5000–10,000 A549 cells were seeded into the wells of a 96-well plate and incubated 48–72 h at 37 °C. Thereafter, the cells were incubated with and without acrolein. The culture medium was changed, 10 μL of 0.05% Cell Counting Kit-8 (CCK-8) reagent (Dojindo Laboratories, Kumamoto, Japan) was added to each well, and the plate was incubated at 37 °C for 3 h. Optical density (OD) was recorded at 450 nm using a microplate reader. Data were computed as a percentage of the indicated groups.

### 4.3. Western Blot Analysis

The membrane and nuclear proteins were extracted using EzSubcell Extract (ATTO, Japan) according to the manufacturer’s instructions. Aliquots of these extracts were separated by 7.5–15% SDS-PAGE and then transblotted onto Immobilon-P membranes (Merck, Germany). After being blocked with 5% skim milk, the blots were incubated with various primary antibodies and then the appropriate secondary antibodies (GE Healthcare Bio-Sciences, Piscataway, NJ, USA). Specific protein bands were detected using ECL Western blotting reagent (GE Healthcare Bio-Sciences, IL, USA), and quantified using a LAS-3000 luminescent image analyzer (Fuji Film, Tokyo, Japan). Antibodies against Na^+^/K^+^-ATPase and c-Jun were obtained from Santa Cruz Biotechnology (Dallas, TX, USA), those against phospho-c-Jun, NF-κB (p-65), phospho-NF-κB (phospho-p-65), HIF-1α, and histone H3 were from Cell Signaling Technology (Danvers, MA, USA), and that against TRPA1 was from Novus Biologicals (Centennial, CO, USA). To measure the level of protein-conjugated acrolein (PC-Acro), cells were lysed in lysis buffer containing 50 mM Tris-HCl (pH 8.0), 150 mM NaCl, 1% Nonidet P-40, 0.5% sodium deoxycholate, 0.1% SDS, and protease inhibitors. The cell lysates were incubated on ice for 30 min with vortexing every 5 min and then centrifuged at 12,000 × *g* for 10 min at 4 °C. Each cell lysate (20 μg protein) was separated on 12% polyacrylamide gel. PC-Acro was detected by an antibody against *N*^ε^-(3-formyl-3,4-dehydropiperidino)-lysine (FDP-lysine) (NOF corporation, Tokyo, Japan).

### 4.4. RNA Extraction and qRT-PCR

Total RNA was extracted from control and acrolein-treated A549 cells using the NucleoSpin^®^ RNA kit (Takara Bio, Gunma, Japan). The mRNA was reverse transcribed to complementary DNA using a cDNA synthesis kit (ReverTra-Plus-^TM^; TOYOBO, Osaka, Japan), which was used as the substrate for quantitative real-time PCR performed using TB Green^®^ Premix Ex Taq^TM^ GC (Takara Bio, Japan) and the Applied Biosystems^®^ 7500 real time PCR system. The primer sequences were as follows: TRPA1 forward primer sequence: 5′-TTCTTGCATTATGCTGCAGCAGAAGGCCAA-3′, TRPA1 reverse primer sequence: 5′-CCTTCATCACCTCATTATTCATGCCCTGCA-3′; β-actin forward primer sequence: 5′-TGGACTTCGAGCAAGAGATGGCCAC-3′, β-actin reverse primer sequence: 5′-CAGACAGCACTGTGTTGGCGTACAG-3′. The expression level of TRPA1 was normalized to that of β-actin.

### 4.5. Transfection

Plasmid transfection was performed using Lipofectamine^TM^ 2000 (Thermo Fisher Scientific, Waltham, MA, USA) according to the manufacturer’s instructions. Expression plasmids (TRPA1: pCMV3-TRPA1; HG17980-UT, vector: pCMV3; CV011) were obtained from Sino Biological (Beijing, China) and knockdown plasmids (pLKO.1-TRPA1 shRNA; TRCN0000044801, vector: pLKO.1-puro; SHC001) were from Sigma-Aldrich (St. Louis, MO, USA). Overexpression and knockdown efficiency were assessed by western blot analysis and cell viability assay, as described above.

### 4.6. Measurement of Polyamines

Cells were homogenized with 5% trichloroacetic acid (TCA) and centrifuged at 12,000 × *g* for 10 min. The supernatant was used for the measurement of polyamines, which were measured as previously described [40], using the Hitachi high-performance liquid chromatography (HPLC) system with a TSKgel Polyaminepak column (4.6 × 50 mm) (Tosoh, Tokyo, Japan) heated to 50 °C. The flow rate of the buffer (0.35 M citric acid buffer, pH 5.1, 2 M NaCl, and 20% methanol) was 0.42 mL/min. Detection of polyamines was by fluorescence intensity with an o-phthalaldehyde solution containing 0.06% o-phthalaldehyde, 0.4 M boric buffer (pH 10.4), 0.1 M Brij-35, and 37 mM 2-mercaptoethanol. The flow rate of the o-phthalaldehyde solution was 0.4 mL/min, and fluorescence was measured at an excitation wavelength of 340 nm and an emission wavelength of 455 nm.

### 4.7. Statistics

Values are indicated as mean ± S.E. of triplicate determinations. Data from control and acrolein-treated groups were analyzed by Student’s *t*-test, and statistical differences were shown by probability values.

## Figures and Tables

**Figure 1 ijms-24-11847-f001:**
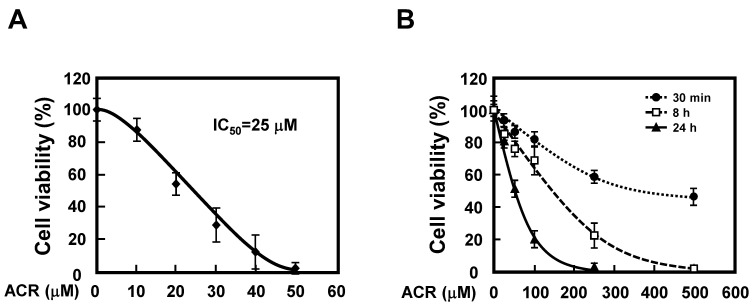
Effect of acrolein on viability of A549 cells. (**A**,**B**) A549 cells were treated with various concentrations of acrolein (ACR) for 72 h (**A**) or different times (**B**). The living cells were then detected using a CCK-8 kit. The cell viability of the control group was set at 100%.

**Figure 2 ijms-24-11847-f002:**
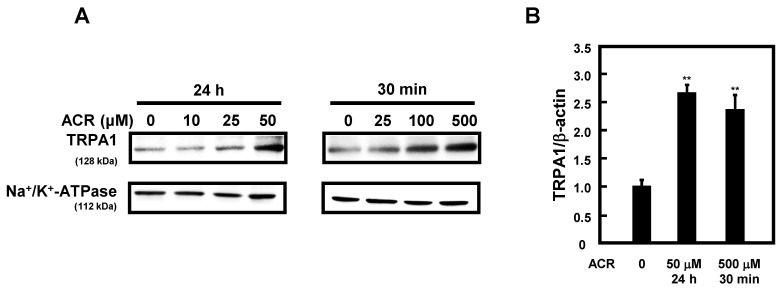
Levels of protein and mRNA of TRPA1. (**A**) TRPA1 in membrane fraction in A549 cells was measured using western blot analysis after incubation of the cells with various concentrations of acrolein (ACR) for 24 h or 30 min. (**B**) The mRNA level of TRPA1 in A549 cells cultured with or without acrolein was determined using qRT-PCR, as described in Materials and Methods. ** *p* < 0.01.

**Figure 3 ijms-24-11847-f003:**
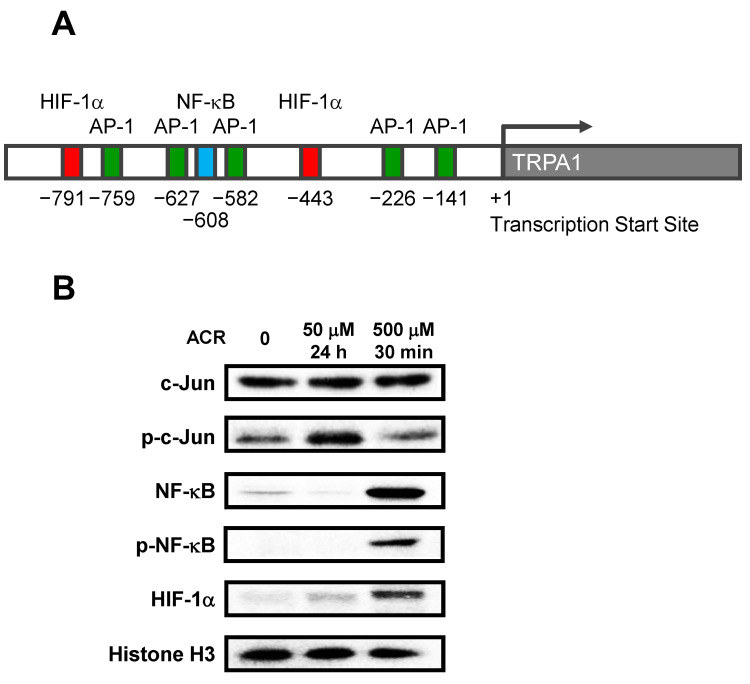
Levels of transcription factors in A549 cells. (**A**) Locations of active sites of transcription factors [AP-1 (c-Jun and c-Fos; green), NF-κB (blue), and HIF-1α (red)] in *TRPA1* gene. (**B**) Levels of the nuclear transcription factors in A549 cells were measured using western blot analysis after incubation of the cells with 50 μM acrolein (ACR) for 24 h or 500 μM acrolein for 30 min. Nuclear histone H3 in A549 cells was used as a control.

**Figure 4 ijms-24-11847-f004:**
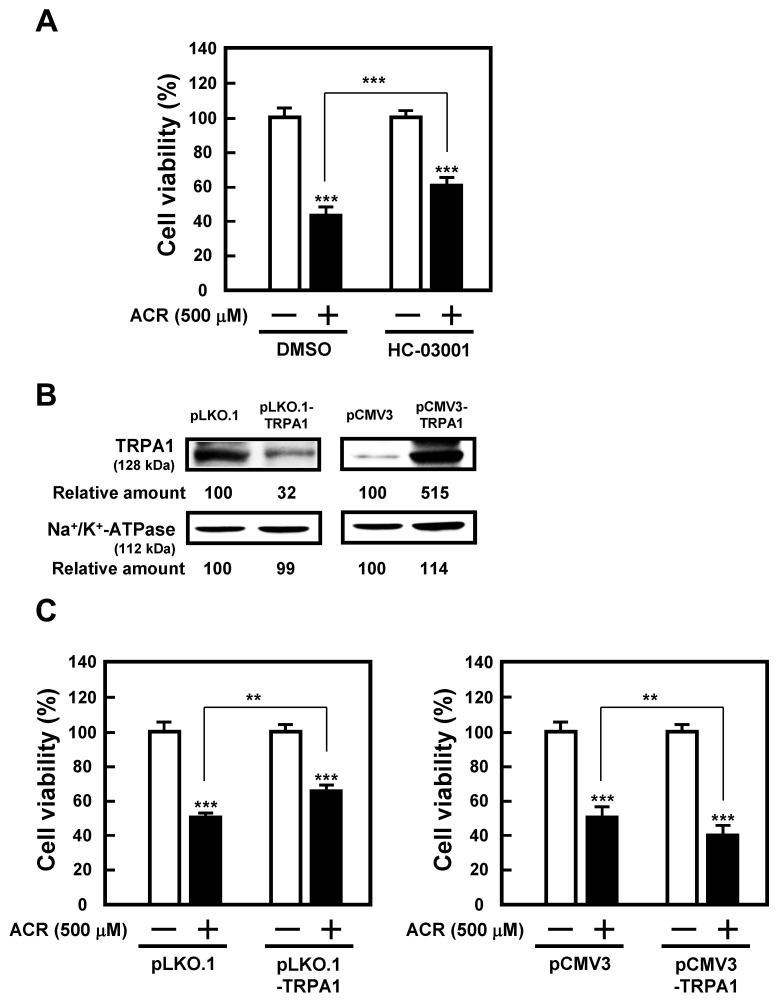
Effect of TRPA1 on acrolein toxicity. (**A**) A549 cells were first preincubated with the TRPA1 antagonist HC-030031 (HC, 100 μM) or vehicle (dimethyl sulfoxide, DMSO) for 30 min. The cells were then exposed to 500 μM acrolein (ACR) for 30 min. Cell viability was measured as described in Materials and Methods. (**B**) Plasmids for knockdown or overexpression of TRPA1 were transfected into A549 cells. Protein levels were analyzed using western blot analysis. (**C**) The cells were treated with 500 μM acrolein for 30 min. Cell viability was measured as described in Materials and Methods. ** *p* < 0.01; *** *p* < 0.001.

**Figure 5 ijms-24-11847-f005:**
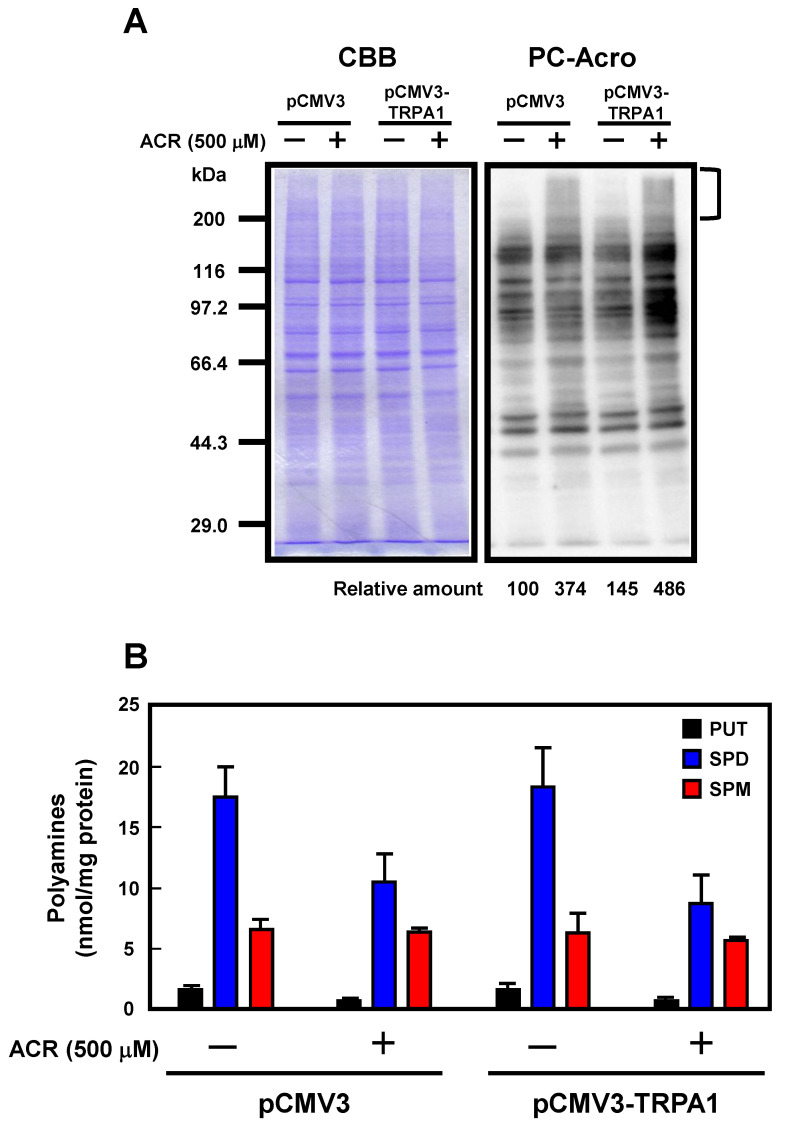
Effect of TRPA1 on the level of PC-Acro in A549 cells. (**A**,**B**) Plasmids for control vector or overexpression of TRPA1 were transfected into A549 cells. These cells were treated with 500 μM acrolein for 30 min. CBB (Coomassie Brilliant Blue) staining and measurement of PC-Acro and polyamines were performed as described in Materials and Methods. The degree of protein polymerization through crosslinking by acrolein (ACR) (shown by square brackets) is indicated as relative amount (%). PUT, putrescine; SPD, spermidine; SPM, spermine.

## Data Availability

All data are contained within the manuscript.

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
