# Peer review of "Transient Receptor Potential Ankyrin 1 (TRPA1) Channel Mediates Acrolein Cytotoxicity in Human Lung Cancer Cells"

_ijms, 2023, doi:10.3390/ijms241411847_

Round 1
Reviewer 1 Report
In my opinion, this manuscript doesnt have any major logical flaws, I believe the design of the study, the setup of the experiments, and their interpretation, are all sound and logical. The choice of tools and cell lines also appears to be okay; at least expression and function of TRPV1 in the cell line under investigation have been tested thoroughly before the experiments were conducted. The conclusions are essentially supported by the data, and the spectrum of different experiments performed (with methods that probably first needed to be optimized) is also without any logical breaks.
One could of course argue a few minor things:
1) what is the concentration of acrolein that can be achieved in bodily fluids, e.g., upon exposure to cigarette smoke? These may be difficult to measure, but it will be beneficial for the document if it is somehow put in the rght dimension. 500 µM of acrolein, which is used for some of the experiments, appears very high to me; I cannot imagin this concentration is ever achieved in vivo.
2) the effects of functionally blocking TRPV1, and its overexpression are mild, to say the least - they may be statistically significant, but they are not prominent. What do you argue may be responsible for this minor effect? That could be discussed.
3) have the results been replicated in at least1 additional cell line? Lung cancer cell lines are an obvious choice, but of course, for validation, using only 1 line is a bit "lower limit". One could also pick lines with differential levels of TRPV1 expression and activity.
Reviewer 2 Report
Please include perspectives and clinical practice implications of these findings. The study will probably have to be replicated in humans.
Can it be a prognostic factor and associated with mortality?
provide more information about TRP gene and about TRP antagonist
I suggest to include the following references, useful for discussion
about smoke compounds, carcinogenesis, HIF, TRP channels
-Future Sci OA. 2019 May 3;5(5):FSO394.
- Pharmaceuticals (Basel). 2018 Sep 21;11(4):90.
-Oncotarget. 2019 Dec 17;10(66):7071-7079.
